# In Vitro Evaluation of the Therapeutic Potential of Phage VA7 against Enterotoxigenic *Bacteroides fragilis* Infection

**DOI:** 10.3390/v13102044

**Published:** 2021-10-11

**Authors:** Nata Bakuradze, Maya Merabishvili, Khatuna Makalatia, Elene Kakabadze, Nino Grdzelishvili, Jeroen Wagemans, Cedric Lood, Irakli Chachua, Mario Vaneechoutte, Rob Lavigne, Jean-Paul Pirnay, Ivane Abiatari, Nina Chanishvili

**Affiliations:** 1Research & Development Department, George Eliava Institute of Bacteriophage, Microbiology and Virology, Tbilisi 0160, Georgia; maia.merabishvili@mil.be (M.M.); khatuna.makalatia@geomedi.edu.ge (K.M.); elene.kakabadze@pha.ge (E.K.); n.grdzelishvili@pha.ge (N.G.); nina.chanishvili@pha.ge (N.C.); 2Department of Biology, Faculty of Exact and Natural Sciences, Javakhishvili Tbilisi State University, Tbilisi 0179, Georgia; 3Laboratory for Molecular and Cellular Technology, Queen Astrid Military Hospital, 1120 Brussels, Belgium; jean-paul.pirnay@mil.be; 4Laboratory Bacteriology Research, Ghent University, 9000 Ghent, Belgium; mario.vaneechoutte@ugent.be; 5Faculty of Medicine, Teaching University Geomedi, Tbilisi 0114, Georgia; 6Institute of Medical and Public Health Research, IIia State University, Tbilisi 0162, Georgia; chachua888@yahoo.com (I.C.); Ivane.Abiatari@iliauni.edu.ge (I.A.); 7Laboratory of Gene Technology, Department of Biosystems, KU Leuven, 3001 Leuven, Belgium; jeroen.wagemans@kuleuven.be (J.W.); cedric.lood@kuleuven.be (C.L.); rob.lavigne@kuleuven.be (R.L.); 8Centre of Microbial and Plant Genetics, Department of Microbial and Molecular Systems, KU Leuven, 3001 Leuven, Belgium; 9School of Medicine, New Vision University, Tbilisi 0159, Georgia

**Keywords:** bacteriophages, phage therapy, enterotoxigenic *Bacteroides fragilis*, ETBF, in vitro model, colorectal carcinoma, CRC, colonic epithelial cells, CEC

## Abstract

Since the beginning of the 20th century, bacteriophages (phages), i.e., viruses that infect bacteria, have been used as antimicrobial agents for treating various infections. Phage preparations targeting a number of bacterial pathogens are still in use in the post-Soviet states and are experiencing a revival in the Western world. However, phages have never been used to treat diseases caused by *Bacteroides fragilis*, the leading agent cultured in anaerobic abscesses and postoperative peritonitis. Enterotoxin-producing strains of *B. fragilis* have been associated with the development of inflammatory diarrhea and colorectal carcinoma. In this study, we evaluated the molecular biosafety and antimicrobial properties of novel phage species vB_BfrS_VA7 (VA7) lysate, as well as its impact on cytokine IL-8 production in an enterotoxigenic *B. fragilis* (ETBF)-infected colonic epithelial cell (CEC) culture model. Compared to untreated infected cells, the addition of phage VA7 to ETBF-infected CECs led to significantly reduced bacterial counts and IL-8 levels. This in vitro study confirms the potential of phage VA7 as an antibacterial agent for use in prophylaxis or in the treatment of *B. fragilis* infections and associated colorectal carcinoma.

## 1. Introduction

Bacteriophages (phages), i.e., viruses that infect bacterial cells, were discovered at the beginning of the 20th century and were immediately used as therapeutic agents to fight bacterial infections. Phages against *Escherichia coli*, *Shigella dysenteriae* and *Vibrio cholerae* were among the first known antimicrobial agents [1]. There have been examples of phage therapy being used effectively to fight various acute and chronic infections [2]. However, we found no such indications in the scientific literature of phage-based treatments of *Bacteroides fragilis* infections including internal organ abscesses, postoperative peritonitis, sepsis, inflammatory diarrhea or colorectal carcinoma (CRC). Species within the genus *Bacteroides*, such as *B. fragilis*, *B. thetaiotaomicron* and *B. vulgatus*, are well-known residents of the natural microbiome of the human and mammalian gastrointestinal (GI) tract mucosa [3]. In these niches, they are responsible for the digestion of cellulose and the production of short-chain fatty acids, while also playing a key role in the maturation of the immune system [4]. Regardless of the numerous beneficial properties, *B. fragilis* also has a pathogenic nature, which is primarily associated with its spread outside the GI lumen following disease, gut perforation, abdominal surgery or trauma. Dislocation of the *Bacteroides* species from the intestinal wall may provoke an inflammatory response and abscess formation in the distant regions of their dissemination, which can be complicated by sepsis [5,6]. Anaerobic abscesses are usually polymicrobial, but one of the most commonly cultured agents is *B. fragilis,* which is suggested to be the most virulent representative of the genus [7]. More specifically, enterotoxigenic *B. fragilis* (ETBF) is associated with inflammatory diarrhea and identified as a risk factor for CRC [8,9,10]. Although antibiotics (e.g., broad-spectrum β-lactams and metronidazole) are generally effective [4], recent studies have shown increased isolation of multidrug-resistant *B. fragilis* strains from the normal microbiome of the human intestines [11].

Increased virulence of *B. fragilis* strains is generally associated with the presence of the *bft* gene in chromosomal pathogenicity island BFT PAI. The *bft* gene codes for fragilysin, a zinc–metalloprotease enterotoxin. Three variants or isoforms of the gene, named *bft-1*, *bft-2* and *bft-3*, have been identified and sequenced, in which *bft-1* was the most prevalent form in humans [12]. This *B. fragilis* toxin (BFT) interacts with epithelial-tight junctions of colonic epithelial cells (CECs) by binding to yet-unidentified receptors [13]. The outcome is the cleavage of intercellular E-cadherin, leading to increased permeability of the epithelial barrier and enhanced activation of the Wnt/β catenin and NF-κB cell-signaling pathways in CECs [14]. In mice, these pathways are shown to induce chemokines to recruit polymorphonuclear immature myeloid cells, with parallel triggering of ETBF-mediated distal colon tumorigenesis [15]. In addition, BFT has been associated with in vitro production of reactive oxygen species and DNA damage—two events linked to carcinogenesis [16]. The *B. fragilis* toxin is also shown to cause increased secretion of IL-8 in human intestinal epithelial cell cultures, which may be a provoking factor for the development of inflammatory diarrhea [14,17]. Cytokine IL-8 is also highly expressed during tumor angiogenesis, leading to the progression of cancer formation [18], and has been found to increase rapidly in ETBF-infected patients [9].

In general, CRC gut microbiota exhibit a compositional shift (dysbiosis), which includes the presence of *B. fragilis*, compared with the microbiota of healthy persons [19]. In 2018, Hannigan and colleagues observed that CRC-associated viromes consisted primarily of temperate phages. They concluded that these phage communities were associated with colorectal cancer and potentially impacted cancer progression by modulating the bacterial host communities towards CRC-associated gut microbiota dysbiosis [20]. Conversely, phages are increasingly considered a promising therapeutic tool against pathogenic gastrointestinal bacteria for their potential to revert dysbiosis of the gut microbiota [21].

In Georgia, sterile bacterial phage lysates, rather than purified phage preparations, are used in oral and topical phage therapies. Since the present study serves to yield preliminary information in view of possible future experimental phage therapies in Georgia, the use of sterile phage lysates was warranted. Our study aimed to assess the potential of a *B. fragilis* phage vB_BfrS_VA7 (VA7) lysate, previously identified as the best candidate for in vitro cell culture experiments [22], in preventing or treating inflammatory ETBF infections and, ultimately, in the prevention of CRC. Previous studies revealed a strictly virulent nature of phage VA7 attributed to *fragilis*-specific phages with a siphovirus morphology (Figure 1). In this preliminary study, we decided to focus on two features of this phage lysate: (a) its antimicrobial properties and (b) its impact on CEC cytokine IL-8 production in an ETBF-infected CEC model.

## 2. Materials and Methods

### 2.1. Identification of ETBF Strains Using PCR

PCR was used to screen thirty *B. fragilis* isolates for the presence of the *bft* gene to help select an adequate ETBF strain for further use in the in vitro studies. Twenty-two *B. fragilis* isolates were obtained from fecal samples of patients with intra-abdominal infections, peritonitis and abscesses [22]. Eight *B. fragilis* strains were obtained from the strain collection at the Laboratory for Bacteriology Research (Ghent University, Ghent, Belgium). DNA was extracted from the bacterial strains using the alkaline lysis method. Briefly, approximately half of a large freshly grown *B. fragilis* colony was suspended in 20 µL alkaline lysis buffer (0.25% SDS, 0.05 N NaOH) and incubated for 15 min at 95 °C. After incubation, samples were centrifuged at 6000× *g* for 5 s and 180 µL of ultrapure water was added to each tube. Samples were centrifuged at the same speed for 5 min to remove the debris. The supernatants were stored for at least 30 min at −20 °C until analysis. PCR-based identification of the *bft* gene was performed according to Aitchison et al. [23]. Forward (5′–3′) GGATACATCAGCTGGGTTGTAG and reverse (5′–3′) GCGAACTCGGTTTATGCAGTGCGAAC primers (Integrated DNA Technologies, Leuven, Belgium) were selected to amplify all three subtypes of the *bft* gene, generating a 296 bp PCR product [20]. PCR master mix was prepared using the FastStart™ High Fidelity PCR System (Roche, Basel, Switzerland), according to the manufacturer’s instructions. PCR cycling conditions were as follows: 15 min of incubation at 95 °C, followed by 35 cycles of denaturation at 95 °C for 30 s, 30 s annealing at 55 °C and 30 s of extension at 72 °C, with a final 2 min extension phase at 72 °C.

### 2.2. Selection and Propagation of an ETBF Host Strain for Phage VA7

A spot test assay was performed to study the activity of phage VA7 against the identified ETBF strains. Bacterial strains were grown in enriched Brain Heart Infusion (BHI) broth (Liofilchem, Roseto degli Abruzzi, Italy). When reaching the exponential growth phase (18 to 24 h after inoculation), bacterial streaks were drawn on Petri dishes with BHI solid medium, using 10 μL loops. Drops of the phage lysates (10 µL) with a titer of 10^8^ PFU/mL were spotted on the air-dried bacterial streaks. The plates were incubated at 37 °C for 18–24 h, after which the visible lytic plaques were evaluated [22]. The productive infectivity of the phages (efficiency of plating (EOP)) was also studied [22]. Determination of the EOP allowed us to define whether phage VA7 was able to replicate in the ETBF strains [24]. One milliliter of 10^9^, 10^8^, 10^7^, 10^6^ and 10^5^ PFU/mL dilutions of each phage was added to 200 μL of 10^8^ CFU/mL of bacteria. After an incubation of up to 10 min at room temperature, 2 mL of semi-solid BHI medium was added, mixed on a shaker and plated on the BHI agar plates. Following overnight incubation, the number of PFUs was counted for each phage dilution. The EOP of each strain was calculated by dividing the number of PFUs formed on the target bacterial strain by the number of PFUs generated by the non-ETBF reference strain A7 [24].

Phage VA7 lysate solutions were prepared by collecting the top layer of double-layer agar plates containing 10^5^ PFU/mL of phages pre-incubated for 24 h with the host bacterial strain A7. The collected suspension was centrifuged at 6000× g for 30 min at 4 °C. The supernatant was filtered through 0.45 µm filters and stored at 4 °C until use. Before applying phages to the cell culture, the lysates were filtered using 0.22 µm filters. The titer of the filtered solution was determined using the double-layer agar method and plated on the BHI solid medium to evaluate the sterility of the phage stocks. The stock phage VA7 lysates produced for the genome sequencing and CEC experiments had a titer of 10^10^ PFU/mL.

### 2.3. DNA Sequencing and Analysis of VA7 Genome

Phage VA7 DNA was extracted from a high-titer phage stock, as described previously [25]. Subsequently, its genome was sequenced using an Illumina MiSeq device (Illumina, San Diego, CA, USA) at the VIB Nucleomics Core (Leuven, Belgium). After sequencing, the raw reads were trimmed (Trimmomatic) and assembled (SPAdes) in one contig [26,27]. Using MEGA X [28], the assembled phage genome was aligned to the closest characterized phage as identified by BLASTn [29] and a ViPTree [30] analysis (Bacteroides phage B124-14; NC_016770). The phage’s taxonomy was further delineated using VIRIDIC [31], which calculates the virus intergenomic distance. Next, the VA7 genome was annotated using RASTtk [32] and manually curated by BLASTp. tRNAs were identified with tRNAscan-SE v2.0. [33,34]. Finally, the phage genome was visualized using EasyFig [34].

### 2.4. One-Step Growth Curve

The phage VA7 one-step growth curve was evaluated to determine the burst size of the infected host cells. The experiment was accomplished according to Kropinski [35]. Bacterial strain E3 was grown in 5 mL of BHI broth for 18 h at 37 °C in a 10% CO_2_ atmosphere. An exponential growth phase E3 culture was diluted in BHI broth (enriched with 1 mM of CaCl_2_) to reach 1 × 10^7^ CFU/mL as a final concentration. The phage VA7 lysate was added to the bacterial culture to attain a multiplicity of infection (MOI) of 0.01 (i.e., the final phage titer was equal to 1 × 10^5^ PFU/mL). The phage–bacterial culture was incubated in a water bath at 37 °C for 10 min to achieve maximal phage to host cell adsorption. Afterwards, 0.1 mL of this mixture was taken to make ten-fold dilutions of up to 1 × 10^1^ PFU/mL. At the same time, 0.01 μL of CHCl_3_ was added to 1 mL of the 1 × 10^2^ PFU/mL dilution to serve as an adsorption control, which was stored on ice till the end of the experiment. The 1 × 10^3^, 1 v 10^2^ and 1 × 10^1^ PFU/mL dilutions of the phage-bacteria mixture were incubated at 37 °C in a water bath for 60 min. Every five to ten minutes 0.1 mL from each dilution was taken to be mixed with 0.2 mL of E3 bacterial culture and 0.6% BHI overlay agar, and poured on a 1.5% BHI solid agar medium. A volume of 0.1 mL of adsorption control was also plated using the same double agar layer method. The plates were incubated for 24 h at 37 °C in a 10% CO_2_ atmosphere. The number of the infected cell and the burst size were determined as described by Kropinski [35].

### 2.5. Effect of Phage VA7 Lysate on ETBF-Infected CEC Cultures

To evaluate the effect of phage VA7 lysate on ETBF-infected CEC cultures, we adapted the model developed by Khan Mirzaei et al. [36]. Colon epithelial cell (CEC) culture line HCT 116, kindly provided by the laboratory of Klinikum rechts der Isar, Technical University of Munich, Germany, was grown in 6-well plates using 10% FCS/DMEM medium (Sigma Aldrich, St. Louis, MA, USA), without addition of antibiotics. After reaching >95% of confluence, CECs were washed twice with PBS buffer and the selected ETBF strain E3 with a final titer of 10^8^ CFU/mL in sterile 10% FCS/DMEM medium was added to each well in a final volume of 3 mL. Blank controls consisted of 3 mL of sterile 10% FCS/DMEM medium without bacteria. For the bacteria to adhere, the CEC cultures were incubated for 3 h at 37 °C in a 5% CO_2_ atmosphere. After incubation, the CEC cultures were washed twice with 1 mL of PBS buffer to remove unattached bacterial cells. Subsequently, 3 mL of phage VA7 lysate (phage stock diluted 1000-fold in 10% FCS/DMEM medium to a final titer of 10^7^ PFU/mL) was applied to the CEC cultures with the already-adhered bacterial cells at an MOI of 0.1. After incubation for another 3 h, the medium was collected and stored at −80 °C for later measurement of IL-8 levels. To remove the CECs from the bottom of the wells, 1 mL of 0.1% of Tween 20 diluted in PBS was added, followed by incubation for 5 min at 37 °C in a 5% CO_2_ environment. Subsequently, the detached cells were diluted ten-fold in the BHI liquid medium and 100 μL was plated on BHI solid medium for bacterial counts. A sandwich ELISA kit was used to measure IL-8 (Abcam, Cambridge, UK) levels, according to the manufacturer’s instructions. Figure 2 summarizes the experimental setup.

### 2.6. Statistical Analysis

The Kruskal–Wallis test was used to evaluate differences in *B. fragilis* cell counts in the various experimental groups, and the one-way analysis of variance (ANOVA), as well as post hoc (Tukey HSD) and Levene’s tests, were run to evaluate differences in IL-8 release levels. The results are presented as mean values (of eight analyses) with error bars representing the 95% confidence intervals (CIs) of the means. We regarded statistical differences to be significant when *p* < 0.05.

## 3. Results and Discussion

Phage VA7 was previously identified as a potential candidate for phage therapy [22]. To evaluate its molecular biosafety, its genome sequence was determined and analyzed. The 47,095 bp double-stranded DNA genome of VA7 has a GC content of 38.53% and shows nucleotide similarity to *Bacteroides* phages B124-14 (96.65% sequence identity, 84% query coverage) [37], Barc2635 (94.80% sequence identity, 87% query coverage), vB_BfrS_23 (94.66% sequence identity, 85% coverage) and B40-8 (94.58% sequence identity, 74% query coverage) [38], all of which are currently unclassified *Siphoviridae* members. A proteome analysis using ViPTree confirmed this result. Next, the virus intergenomic distance between the most related phages and VA7 was calculated (Appendix A), showing that *Bacteroides* phages VA7, B124-14, Barc2635, vB_BfrS_23 and B40-8 form five novel species within one novel unclassified *Siphoviridae* genus according to the International Committee for the Taxonomy of Viruses (ICTV) guidelines [39,40].

Next, the VA7 genome was annotated, revealing 68 coding sequences and no predicted tRNAs (Figure 3). The annotated genome was submitted to NCBI and is available through Genbank accession number MW916539.1. No lysogeny-related proteins are encoded on the VA7 genome, indicating its strictly lytic character. Moreover, no known virulence- or antibiotic resistance-associated genes were identified, suggesting that phage VA7 could be considered as a suitable candidate for phage therapy purposes, even though the majority of the coding sequences (72%) remain of unknown function.

The phage VA7 one-step growth curve showed a latent period of about 20 min and a burst size at 30 min of about three hundred particles per infected bacterial cell (Figure 4).

To identify the enterotoxigenic *B. fragilis* strains in our previously described collection [22], the thirty available strains were analyzed for the presence of the *bft* gene by a PCR assay. This led to the amplification of a *bft* gene fragment with a correct size (296 bp) in four out of the thirty isolates: E1, E3, S10 and 33.

Among these, ETBF strain E3 was shown to be the most suitable host for phage VA7, being susceptible to the phage in a spot test assay and exhibiting an EOP of 0.1, which is indicative of an efficient production of new VA7 virions (productive infection) in the E3 strain. The virulence potential of strain E3 towards CECs was evaluated using a cytotoxicity assay (Appendix A). Although ETBF strain S10 had an EOP similar to E3, the latter produced clearer lytic plaques. Phage VA7 showed lytic activity on ETBF strains E1 and 33, but no propagation was observed [22]. Therefore, the E3 strain showing susceptibility to phage VA7 was selected as a test strain in the cell culture experiments, as well as for the propagation of phage lysates used in the cell culture experiments.

Phage VA7 lysate was then evaluated for its antibacterial properties on ETBF-infected CEC cultures. Here, we observed a statistically significant (*p* < 0.05) decrease in bacterial cell counts in the phage-treated arm compared to the untreated infected cell culture arm. The mean bacterial cell count in the untreated ETBF (strain E3)-infected arm was 5 × 10^4^ CFU/mL, while the mean bacterial count in the phage VA7 lysate-treated arm was about 2 log units lower (4.2 × 10^2^ CFU/mL) (Figure 5a).

The immunomodulating effect of VA7 was also investigated in the CEC cultures. A statistically significant difference in IL-8 concentrations was observed between the phage VA7 lysate-treated arm and the untreated arm. With a mean level of 391.99 pg/mL, IL-8 production was significantly lower (*p* < 0.001) in the phage arm as compared to the untreated arm (514.74 pg/mL) (Figure 5b).

To the best of our knowledge, there are no reports in the scientific literature of the application of phages for the treatment of *B. fragilis* infections or the in vitro assessment of their therapeutic potential. Previously well-studied *B. fragilis*-specific phages HSP-40 and GB-124-14 were used only for the monitoring of fecal contamination in different water environments [41,42].

A previous study identified phage VA7 as a potential candidate for application against ETBF infections within in vitro or animal models, or in human phage therapy [22]. In the current study, we sequenced the genome of VA7, further supporting that it is a strictly virulent phage and that there are no counter-indications for its use in human phage therapy.

We also evaluated the antimicrobial properties of phage VA7 lysate and its impact on CEC cytokine IL-8 production in an ETBF (strain E3)-infected CEC model. Similar studies were performed on the phages of other pathogens such as *E. coli*, *Pseudomonas aeruginosa* or *Staphylococcus aureus* [43,44]. We observed that the addition of phage VA7 lysate to E3-infected CECs led to significantly reduced bacterial counts and lower IL-8 levels compared to untreated infected cells. These lower IL-8 levels could be explained by a potential decrease in BFT production, which, in turn, is expected to be linked to the reduction of bacterial counts [45]. Previous studies showed that a reduction in the bacterial load by cefoxitin led to the regression and disappearance of ETBF-associated neoplastic formations in the colon of mice [46]. A decrease in IL-17A, which is a pro-inflammatory cytokine released by Th-17 lymphocytes and found to be highly involved in cancer progression, was also observed [47]. By the activation of NF-κB and p42/44MAPK pathways, IL-17A was shown to stimulate the production of cytokine IL-8 [47]. IL-8 is involved in the angiogenesis of developing tumors. We speculate that directly, via BTF release, or indirectly through IL-17A stimulation, IL-8 levels are increased in the presence of ETBF and that both cytokines’ levels can be reduced through the elimination of the pathogenic cells. Even though our results are based on in vitro experiments, we suggest that the reduction in bacterial count caused by VA7 may prevent or even reverse the formation of colorectal adenocarcinoma caused by ETBF.

In the country of Georgia, phage preparations for oral and local administration are mainly produced and distributed in their lysate forms. Subsequently, we deliberately neglected to purify the phages before using them in the present model. The goal was to evaluate the antibacterial activity and CEC IL-8 response of the phage formulations that would ultimately be applied in future human prophylaxis or therapy applications in Georgia. However, the tested VA7 phage lysates undeniably contained BTF due to the production of the phage lysate in ETBF strain E3. Even though this BFT level is bound to be very low, as the lysates were diluted 1000-fold (from 10^10^ to 10^7^ PFU/mL) in cell culture medium before addition to the present in vitro model, it is bound to have generated a certain bias.

We acknowledge that these are preliminary results. Future studies to evaluate the impact of phage VA7 on the induction of other relevant chemokines such as IL-17 and NF-κB, and the determination of CEC viability, are envisioned. Relevant biophysical characteristics of phage VA7, such as its stability in the human body or various solutions, and its penetration and distribution ability in relevant bodily tissues, are also of interest. These future pre-clinical studies will also include the evaluation of purified phage VA7 preparations.

## Figures and Tables

**Figure 1 viruses-13-02044-f001:**
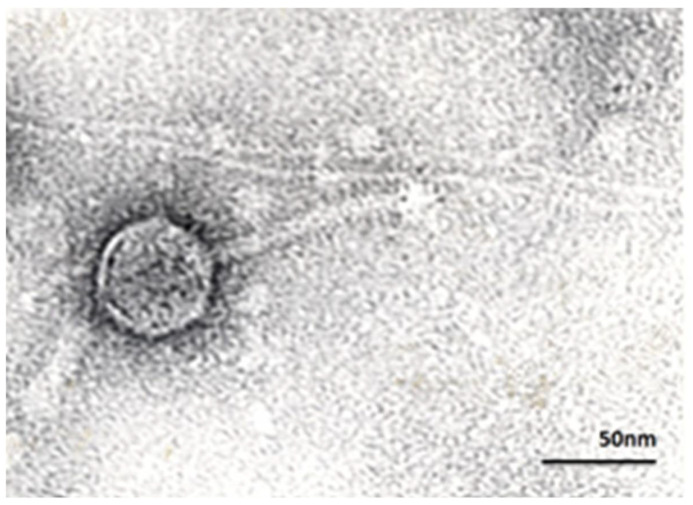
Transmission electron micrograph of *B. fragilis*-specific phage VA7, exhibiting a siphovirus morphology with an icosahedral head of 60 nm and a tail of 100 nm. Magnification × 250,000.

**Figure 2 viruses-13-02044-f002:**
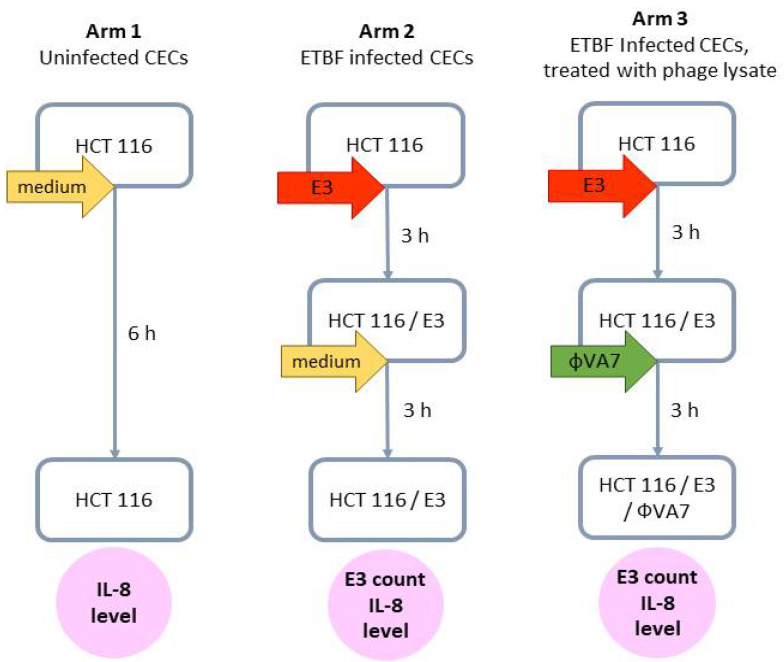
Scheme of the experimental design to evaluate the effect of phage VA7 lysate on enterotoxigenic *B. fragilis* (ETBF) infection and IL-8 production in colonic epithelial cell (CEC) cultures. For each arm, two distinct experiments (biological replicates) were performed, and for each of these, four samples (technical replicates) were analyzed. In other words, each data point was the result of eight analyses.

**Figure 3 viruses-13-02044-f003:**
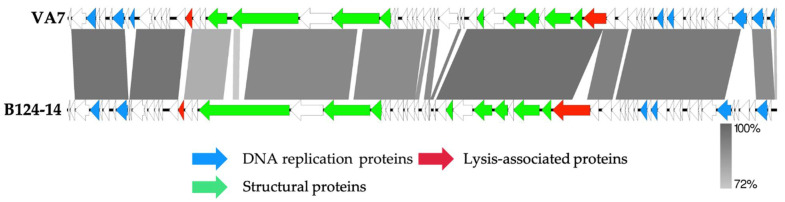
Genome map of the sequenced *Bacteroides* phage VA7 and comparison using a BLASTn analysis (greyscale) to the closest related phage B124-14. Each arrow represents a coding sequence. In red, genes encoding packaging and lysis-associated proteins are displayed, while green shows the structural proteins and blue the DNA- and metabolism-associated proteins (adapted from EasyFig).

**Figure 4 viruses-13-02044-f004:**
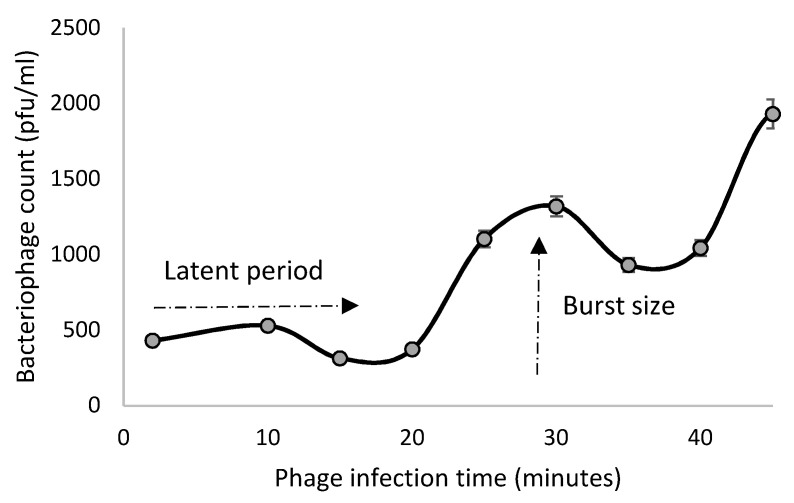
One-step growth curve of phage VA7 in bacterial host A7. The evolution of phage lytic activity (PFU/mL) over time (min) is shown. The data points represent the mean of three experiments with the error bars.

**Figure 5 viruses-13-02044-f005:**
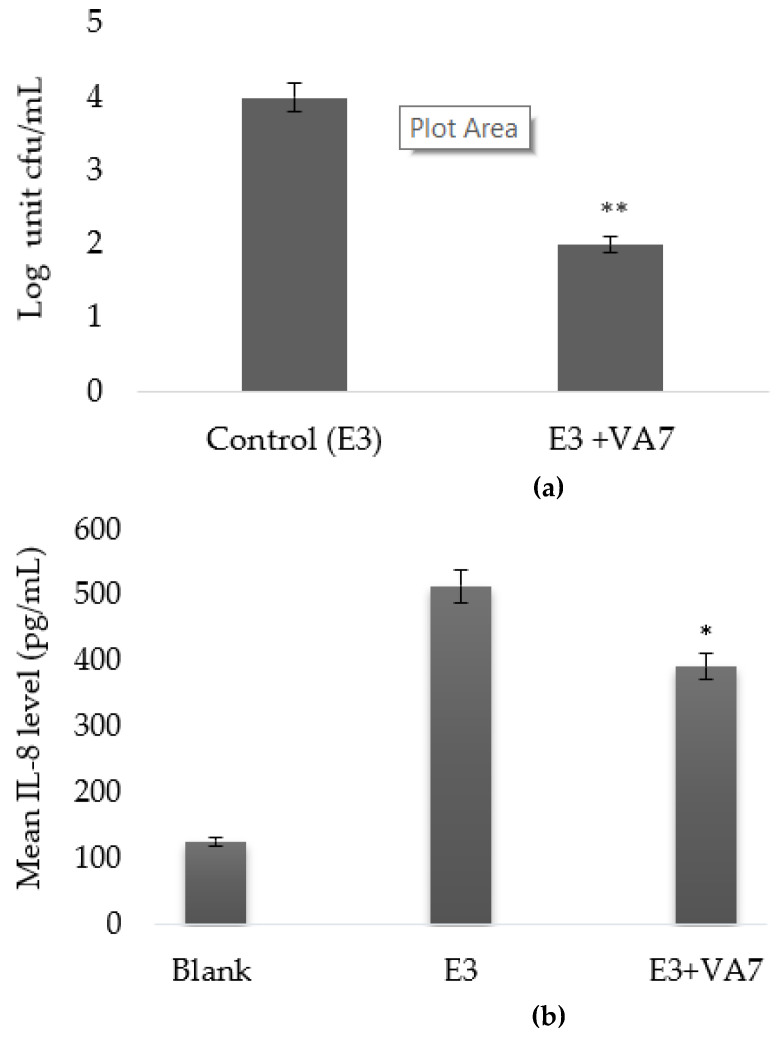
Bactericidal effect (**a**) and IL-8 modulating effect (**b**) of VA7 lysate on HCT116 colonic epithelial cell (CEC) cultures infected with enterotoxigenic *B. fragilis* (ETBF) strain E3. Bacterial counts and IL-8 levels are presented as mean values with error bars representing the 95% confidence intervals (CIs) of the means. A base-10 logarithmic scale is used for the Y-axis of graph a. A statistically significant difference is indicated as * *p* < 0.05 and ** *p <* 0.01.

## Data Availability

Data is contained within the article or Appendix A. The annotated genome was submitted to NCBI and is available through Genbank accession number MW916539.1.

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
