# Peer review of "In Vitro Evaluation of the Therapeutic Potential of Phage VA7 against Enterotoxigenic *Bacteroides fragilis* Infection"

_viruses, 2021, doi:10.3390/v13102044_

Round 1

Reviewer 1 Report

Dear authors,

In the present study “In vitro evaluation of the therapeutic potential of a phage VA7 against enterotoxigenic Bacteroides fragilis infection” by Bakuradze et al., the authors describe a novel phage species vB_Bfr_VA7 (VA7) for potential use as an antimicrobial agent. The phage is envisaged to treat diseases caused by Bacteroides fragilis, the leading agent cultured in anaerobic abscesses and postoperative peritonitis. Enterotoxin-producing strains of B. fragilis have been associated with the development of inflammatory diarrhea and colorectal carcinoma. The authors evaluated antimicrobial properties of the phage lysate, and its impact on cytokine IL-8 production in an enterotoxigenic B. fragilis infected colonic epithelial cell culture. Here, the addition of phage VA7 led to reduced bacterial counts and IL-8 levels.

The manuscript deals with a societal important problem, which is about combating serious infections with antibiotic alternatives. Phages can be an efficient and effective tool to eradicate bacterial infections. Although, several researchers already focus on phage therapy, “uncommon” pathogens, such as Bacteroides fragilis, are mostly neglected. is nicely written. Thus, studies like the one presented here are of great societal and scientific importance. The manuscript is scientifically sound. In addition, the abstract appropriately summarizes the study. The introduction covers all necessary points to understand the following results and methods. The results section is clearly presented, and the results mostly support the discussion/conclusion. 

Comments and questions of interest are implemented in the attached pdf file.

Author Response

Please find a PDF file with the response.

Reviewer 2 Report

The manuscript of Bakuradze et al. deals with sequencing and characterization of a Bacteroides fragilis-specific bacteriophage. Despite the whole material is short, the study design and implementation were sound and the standard of communication is high. However, additionally to those that authors mention there are some more open questions that should be taken into consideration, e.g. what is the species-specificity of the VA7 phage, why the authors chose only ETBF strains, what would be route of its administration and how it would affect the intestinal microbiome. Without these, the manuscript is fairly theoretical.

Specific comment

Figure 3, blue arrow in the legend: in spite of "DNA-associated proteins" use  more appropriate expression(s), e.g. like replication, recombination, transcription.  

Author Response

(The authors gave the same response as above.)

Round 2

Reviewer 2 Report

The authors answered my questions which makes the manuscript acceptable.